# Modelling the Reliability of Logistics Flows in a Complex Production System

Bożena Zwolińska and Jakub Wiercioch *

Faculty of Mechanical Engineering and Robotics, AGH University of Krakow, 30-059 Krakow, Poland; bzwol@agh.edu.pl
* Correspondence: wiercioc@agh.edu.pl; Tel.: +48-607-940-770

**Abstract:** This paper analyses the disruptions occurring in a production system determining the operating states of a single machine. A system with a convergent production character, in which both single flows (streams) and multi-stream flows occur, was considered. In this paper, a two-level formalisation of the production system (PS) was made according to complex systems theory. The continuity analysis was performed at the operational level (manufacturing machine level). The definition of the $k$th survival value and the quasi-coherence property defined on chains of synchronous relations were used to determine the impact of interruption of the processed material flow on uninterrupted machine operation. The developed methodology is presented in terms of shaping the energy efficiency of technical objects with the highest power demand (the furnace of an automatic paint shop and the furnace of a glass tempering line were taken into consideration). The proposed methodology is used to optimise energy consumption in complex production structures. The model presented is utilitarian in nature—it can be applied to any technical system where there is randomness of task execution times and randomness of unplanned events. This paper considers the case in which two mutually independent random variables determining the duration of correct operation $T_P$ and the duration of breakdown $T_B$ are determined by a given distribution: Gaussian and Gamma family distributions (including combinations of exponential and Erlang distributions). A formalised methodology is also developed to determine the stability of system operation, as well as to assess the potential risk for arbitrary system evaluation parameters.

**Keywords:** energy consumption optimisation; reliability modelling; exploitation process modelling; fault models; maintenance; logistics; lean production; agile manufacturing; industry 4.0; $k$th survival value

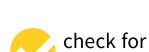



## 1. Introduction

In order to maximise adaptation to the changing consumer environment, companies need to follow environmental and legal changes. A major determinant of success is the ability to adapt to a changing market environment [1,2]. This characteristic is defined as flexibility [3]. Flexibility of organisations can be achieved in various aspects of the business [4–6]. In real manufacturing systems, flexibility at the operational level is achieved through the use of multi-purpose machines and production control algorithms [7,8]. The use of lean production and agile manufacturing paradigms supports production processes in companies by contributing to their flexibility and economic efficiency, as well as reducing waste [9–12]. Lean production and agile manufacturing can be combined with the use of Industry 4.0 technologies [13–18]. This facilitates the preservation or implementation of cost-competitive production or increased efficiency in digitised manufacturing companies and so-called smart factories of the Industry 4.0 era [15,19–22]. This applies to large global companies, as well as SMEs (small and medium-sized enterprises) [23,24], at different levels of the processed material and supply chain [25,26]. In lean manufacturing, companies aim to increase economic efficiency, improve product quality and minimise lead times [27–30].

The current trend is to use lean manufacturing tools not only for economic but also for environmental benefits [31]. Similar to the agile manufacturing concept, the lean production concept also has identified difficulties in implementing and integrating into Industry 4.0 [9,16,32,33]. These include management difficulties, lack of strategy for implementing Industry 4.0, risk of lack of proper cyber security, digital skills, and staff resistance [34–37].

In order to achieve a greater ability to react to rapidly changing customer requirements and preferences, agile manufacturing is defined as a business-wide mindset, characterised by a significant emphasis on flexible structures and increased access to global competencies [38–41]. Operational-level agile cannot be generalised to individual companies [42]. The challenges and difficulties in implementing agile manufacturing have been identified in the literature as team motivation, team collaboration, conflict resolution, building an agile mindset, technological, organisational, and environmental [43]. In agile manufacturing, the controlled pace of manufacturing can be adapted to current needs [44]. The production schedule of products depends on the company's situation [45]. The flexibility of the production system should be assessed for sustainable development considering social, economic, and ecological aspects [46–48]. Modern companies give equal consideration to technical–technological development and operation in line with zero waste or zero emissions [49–51]. Currently, an important functional and organisational aspect of enterprises is the policy of maximum carbon dioxide emission reduction [52]. The introduction of regulations related to the $CO_2$ emissions trading system is the foundation of an effective tool to control and minimise emissions on a national and international scale [53]. Significant zero-carbon results are achieved by maximising the use of renewable energy sources, such as solar, wind, hydro, and/or geothermal energy [46,54]. The implementation of a zero-carbon programme in manufacturing companies is multi-phase and multi-area [54,55]. In the first investment phase, the energy demand is determined, taking into account the potential development on a strategic scale (several years) [50]. Potential cost-saving programmes are also evaluated at this stage. One such solution is a holistic assessment of the impact of the machinery failure rates on energy consumption. Complex manufacturing systems achieve their greatest production efficiency when the machinery is at maximum capacity [56]. Each technical object is characterised by its own lifespan and failure rate [56,57]. Lean production includes methods for the effective management of maintenance systems in order to maximise efficiency while minimising costs [58]. The accuracy of the performance improvement tools used is increased by the application of mathematical, statistical, and stochastic knowledge. The use of extensive algorithms taking into account machine life functions makes it possible to model a real object and then test its behaviour a priori. Consequently, this makes it possible to verify the developed model without increasing the risk of the implementation of ineffective solutions.

In the general design assumptions, the energy demand is determined for the maximum power needed, with an assumption of ±20% fluctuations potentially occurring [46]. Systems with a higher level of risk have greater safeguards [46], sometimes up to 400% [59]. The unjustified oversizing of the energy demand is associated with unreasonable costs for the investment phase. The management of energy resources during operation is characterised by a different range of activities. This paper presents a model, whose aim is to support optimal decision-making regarding the energy consumption of equipment with the highest power demand. The main assumption of the developed model is to minimise the idle operation energy losses of the machines: the automatic paint shop furnace and the glass tempering furnace. In the analyses, failures causing disruptions in material flows were considered. On the basis of the analyses of the real object, a model was developed for a production system (PS) with a series-parallel structure. Although this presented methodology refers to a specific object, it has utilitarian properties. After adjusting the flow structure, the presented algorithm can be applied to any manufacturing system.

This proposed methodology concerns the assessment of the impact of the downtime of a single machine on the entire production system. In the scope of this article, the model was exemplified for separated subsystems where the relation chains supply the machines with

the highest energy demand with the processed material. These machines are the furnace in the automatic paint shop and the glass tempering furnace. Technical objects with complex material flows, where linear and multi-stream flows can be distinguished, were chosen for the analysis. Moreover, different flow management systems can be distinguished in each of these cases, either of the pull or push type. The complexity of the analysis is also influenced by the dichotomy in the processed material flows of the automatic paint shop furnace. These aspects are taken into account in the developed algorithm.

The essence of the presented model is the Total Productive Maintenance (TPM) method's failure assessment indicators—Mean Time To Failure (MTTF) and Mean Time To Repair (MTTR) [28,60–63]. The model presented in this paper takes into account the randomness of events occurring in the real system. The purpose of the developed algorithm is to reduce the energy consumption losses of machines with the highest energy demand. Hence, for separated material flows, an analysis was made of the impact of failures on objects belonging to chains of synchronous relations supplying the furnace of the automatic paint shop and the furnace of the glass tempering line. This presented methodology can be also applied to determine the stability of the operation of a complex system and assess potential risks.

## 2. Materials and Methods

In this article, a complex production system with a convergent manufacturing character (Figure 1) is analysed. Production system complexity is a property that affects the difficulty of defining and modelling it, the difficulty of understanding it, and the difficulty of using it—commonly, the complexity of a production system is the difficulty of understanding, using, and managing it. According to complex systems theory, a complex system is considered to be a system with a relatively high degree of structural complexity, where, after decomposition, subsystems that are extracted can be considered complex systems in further analyses, and/or in overarching terms, and there is a high degree of complexity in the interrelationships between the separated subsystems. The model of the object chosen for analysis takes into account a two-level decomposition of the PS components with defined chains of relations at each level of decomposition. Hence,

$$PS = \{E, R\}, \tag{1}$$

where: $PS$—production system; $R$—relations between the $PS$ elements; and $E$—the elements belonging to the $PS$ separated at the first level of decomposition, which are a set of departments $D_d$, for $d = 1, 2, \ldots, D$:

$$E = \{D_1, D_2, \ldots, D_d\}. \tag{2}$$

The second level of decomposition of the system consists of machines $M_{m(d)}$ for $m = 1, 2, \ldots, M$ belonging to a specific department $D_d$ performing a given production process. Then,

$$D_d = \left\{ M_{1(d)}, M_{2(d)}, \ldots, M_{M(d)} \right\}, \tag{3}$$

where $M_{m(d)}$ determines the machine with the number $m$ in the department with the number $d$ for $m = 1, 2, \ldots, M$ and $d = 1, 2, \ldots, D$.

The technical–technological solutions of the system define the strict sequence of relations existing in a given production system [64]. The machine park determines the production technology, on the basis of which the so-called technological route is formalized. In a series-parallel production structure, a specific production process can be executed on a single machine or on a number of machines of technological similarity. Hence, for a given technological route, a scheduling route is defined. A scheduling route is the strict assignment of specific machines to a defined technological route. A sequence of production processes performed using machines $M_{m(d)}$ defines the flow relations of processed materials in the PS. In the case under consideration, relations at two levels

of decomposition can be distinguished. The first level is the sets of relations $R^D \subset R$ existing between the departments $D_d : \ \forall \ d = 1, 2, \ldots, \ D$, which are defined on the basis of the technological route. The second level of relations is defined on the basis of the scheduling route; these are sets of relations $R^M \subset R$ existing between machines $M_{m(d)} : \ \forall m(d) = 1(d), 2(d), \ldots, \ M(d) \ \vee \ \forall d = 1, \ 2, \ldots, D$. Then,

$$R = \left\{ R^D, R^M \right\}. \tag{4}$$

Analyses of $R^D \subset R$ relations refer to assessments of system performance at the tactical level of the company. Operational-level performance evaluations refer to the $R^M \subset R$ relations existing between machines $M_{m(d)}$. For the case under consideration, the set of $R^D$ relations is defined as follows:

$$R^D = \left\{ \mathcal{R}_{r(d,\hat{d})} : \ d, \hat{d} \in \{1, 2, .., D\} \ \wedge \ d \neq \hat{d} \right\}. \tag{5}$$

The set of $R^M$ relations is defined as follows:

$$R^M = \left\{ \mathcal{R}_{r(m_i(d), m_j(d))} : \ i, j = 1, 2, \ldots, M(d) \ \wedge \ i \neq j \ \wedge d = 1, \ 2, .., D \right\}, \tag{6}$$

where $\mathcal{R}_{r(m_i(d), m_j(d))}$ determines the relation numbered $r$ for $r = 1, 2, \ldots, \check{\mathcal{R}}$.

The relations $R \subset PS$ are the set of relations $\mathcal{R}_{r(d,\hat{d})}$ and $\mathcal{R}_{r(m_i(d), m_j(d))}$. The set of relations $\mathcal{R}_{r(d,\hat{d})}$ exists between departments numbered $d$ and $\hat{d}$ for $d, \hat{d} = 1, 2, \ldots, D$. Meanwhile, the set of relations $\mathcal{R}_{r(m_i(d), m_j(d))}$ exists between machines numbered $m_i$ and $m_j$ for $i, j = 1, 2, \ldots, M(d)$. The sets of relations $\mathcal{R}_{r(d,\hat{d})}$ and $\mathcal{R}_{r(m_i(d), m_j(d))}$ are considered separately at the appropriate decomposition level. Thus, the numerator $r = 1, 2, \ldots, \check{\mathcal{R}}$ varies depending on the considered sets $R^D$ or $R^M$ (relations $\mathcal{R}_{r(d,\hat{d})}$ and $\mathcal{R}_{r(m_i(d), m_j(d))}$ are not considered simultaneously).

For convergent systems, the sequences of relations $\mathcal{R}_{r(m_i(d), m_j(d))} \subset R^M$ illustrate the actual flow of materials through the various production processes. In a given flow, the machine $M_{m_i(d)}$ is the origin of the relation $\mathcal{R}_{r(m_i(d), m_j(d))}$. In turn, the machine $M_{m_j(d)}$ is simultaneously the end of the relation $\mathcal{R}_{r(m_i(d), m_j(d))}$ and the beginning of the relation $\mathcal{R}_{r+1(m_i(d), m_j(d))}$. The relation chain is composed of a sequence of multiple relations $\mathcal{R}_{r(m_i(d), m_j(d))}$ for $r \in \left\{ 1, \ldots, \check{\mathcal{R}} \right\}$ and can form adequately long transition paths that depend on the length of the processed material chain. If there is no disruption in the processing of the selected component of the bill of materials (BOM) structure during the time interval $\Delta t$, a specific sequence of single synchronous interactions defined as a chain of $^{r(M_{i(d)}, M_{j(d)})}\mathcal{R}$ is created.

In any production system, there are synchronous and asynchronous relations. Depending on the type of analysis performed, synchronous and asynchronous relations can be considered at the tactical level for relations $\mathcal{R}_{r(d,\hat{d})} \subset R^D$ and at the operational level for relations $\mathcal{R}_{r(m_i(d), m_j(d))} \subset R^M$. In the scope of this article, flows between machines are considered; hence, the formalisation of synchronous and asynchronous relations is defined for the set of relations $R^M \subset R$.

A synchronous relation is a relation $\mathcal{R}_{l(m_i(d), m_j(d))} \subset R^M$ ($l \in \left\{ 1, 2, \ldots, \check{\mathcal{R}} \right\}$) existing between machines $M_{i(d)}$ and $M_{j(d)}$ for $i \neq j$ and $d = 1, 2, \ldots, D$, where $i \in \{1(d), 2(d), \ldots, M(d)\}$ and $j \in \{1(d), 2(d), \ldots, M(d)\}$ if, during the operation of the production system, there exists such a timeframe $[t_1, t_T]$ or a sequence of times $t_1 \leq t_2 \leq \ldots \leq t_{T-1} \leq t_T$ in which $\mathcal{R}_{l(m_i(d), m_j(d))}$ occurs continuously. A relation $\mathcal{R}_l$ is a synchronous relation existing cyclically when, for an existing sequence of times $t_1 \leq t_2 \leq \ldots \leq t_{T-1} \leq t_T$ inflicting a division of the timeframe $[t_1, t_T]$ into sections of equal length $[t_k, t_{k+1}]$ for $k = 1, 2, \ldots, T - 1$, a subsequence of timeframes can be distinguished in which this rela-

tion exists continuously and uninterruptedly. Asynchronous relations exist when it is not possible to explicitly define a timeframe or a subsequence of times.

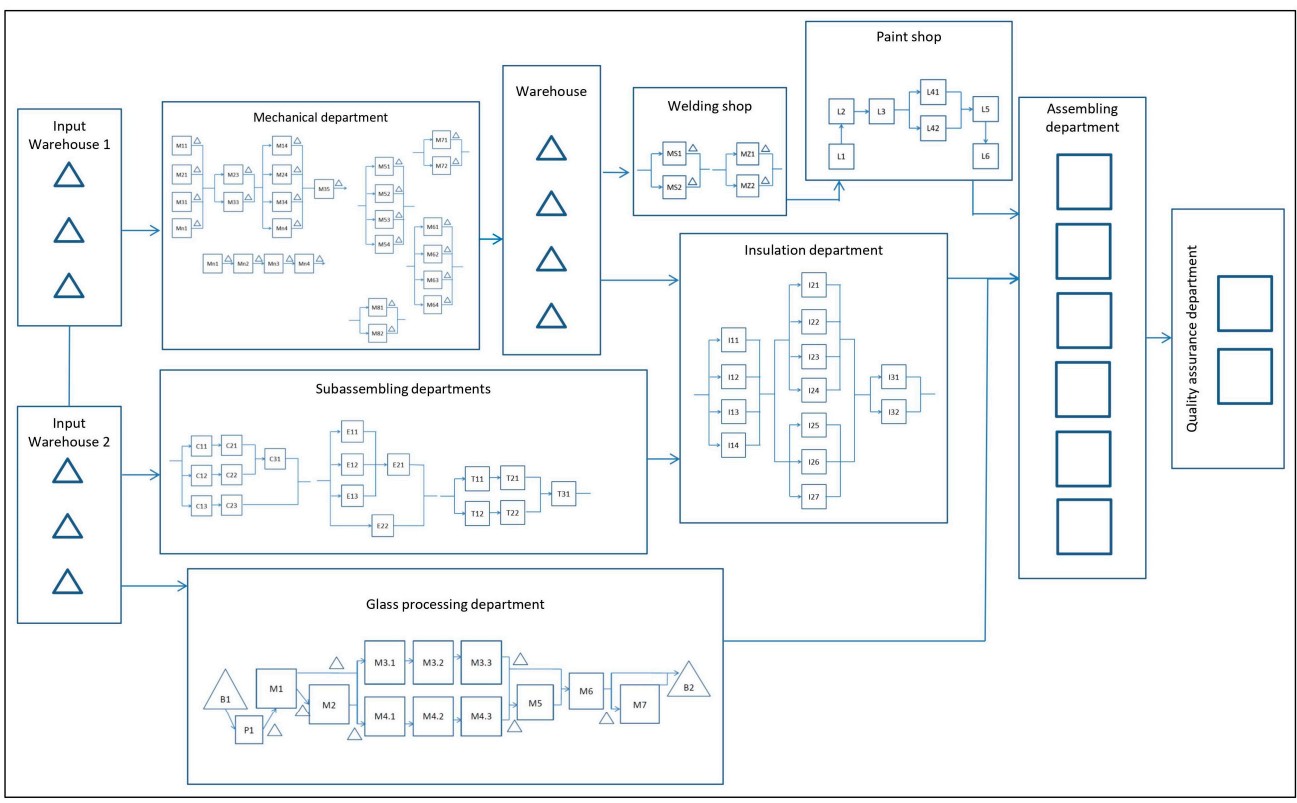

**Figure 1.** Scheme of the series-parallel production structure chosen for the analysis.

For any given production structure (linear or series-parallel), properly executed production processes form a sequence (or sequences) of relations $\mathcal{R}_{r(m_i(d),m_j(d))}$. Following the successive stages of production of an arbitrarily selected component, a specific chain of relations defined by a scheduling route will be created. Hence, a chain of relations $^{r(M_{i(d)},M_{j(d)})}\mathcal{R}$ is called a defined set of relations $\mathcal{R}_{r(m_i(d),m_j(d))}$ with a strictly assigned order. The existing chains of relations at any $\Delta t$ may be either synchronous or asynchronous. A synchronous relation chain is a sequence of single relations that in the time $[t_1, t_T]$ or in the time subsequences $t_1 \leq t_2 \leq \ldots \leq t_{T-1} \leq t_T$ are synchronous ($\mathcal{R}_{r(m_i(d),m_j(d))} \in {}^{r(M_{i(d)},M_{j(d)})}\mathcal{R}$). If there is at least one asynchronous relation in a chain of relations $^{r(M_{i(d)},M_{j(d)})}\mathcal{R}$ at a given $\Delta t$, then the chain is asynchronous. The asynchronicity of a production system is determined by disruptions in the flow of material being processed. In lean production, these are disruptions resulting from the occurrence of muri, mura, and/or muda wastage [65].

Muri, mura, and muda disruptions result in the occurrence of system variability and loss of production system stability. Stability of the production system refers to the ability to keep the value of the evaluated parameter in equilibrium. In the area of the production system, all processes are stochastic in nature. Hence, a PS is considered stable if the values of the evaluated characteristic at a given $\Delta t$ are within $\pm 3\sigma$. This variability is referred to as natural variability. In case of undesirable random events, there are disturbances that can cause the value of the assessed characteristic to exceed the $\pm 3\sigma$ limits. This variability is referred to as special. A production system is considered stable if it has the ability to return to equilibrium after a disruption, preserving the functionality of the operation. This paper considers disruptions that interrupt the flow of processed materials and are caused by unplanned downtime of machines on the scheduling route.

In this paper, the quasi-coherence property of a set of synchronous relations is used to determine the effect of disruptions on the operation of a specific object. Quasi-coherence

is defined as the correct (undisturbed) flow of processed material between any machines $M_{i(d)}, M_{j(d)} \subset D_d$ for $i \neq j \wedge d = 1, 2, \dots, D$. Quasi-coherence analyses can be performed on single relations $\mathcal{R}_{r(m_i(d), m_j(d))}$ or chains of relations $^{r(M_{i(d)}, M_{j(d)})}\mathcal{R}$. The term quasi-coherence at the time $\Delta t = [t_1, t_T]$ is used to describe the undisturbed material flow resulting from correctly executed production processes. In the case of chains of relations at a given time sequence $t_1 \leq t_2 \leq \dots \leq t_{T-1} \leq t_T$, the so-called semi-quasi-coherence may occur. The semi-quasi-coherence is considered when it is possible to extract an uninterrupted flow of processed material at certain time periods with length $[t_k, t_{k+1}]$ for $k = 1, 2, \dots, T-1$ in a subset of single relations $\mathcal{R}_{r(m_i(d), m_j(d))}$ belonging to a defined relation chain $^{r(M_{i(d)}, M_{j(d)})}\mathcal{R}$. For the semi-quasi-coherence case, there is no uninterrupted material flow in the entire relations chain but only in its individual parts. In the analysis of relations chains in linear flows without inter-operational buffers, only two cases are possible: (1) the system has the quasi-coherence property, or (2) the system does not have the quasi-coherence property. One of the straightforward solutions to improve the continuity of the flow of material being processed is the introduction of inter-operational buffers. Then, three operating states can occur in a linear system, such as in a combined (series-parallel—Figure 1) structure. In considering series-parallel systems with convergent or divergent production, three types of system operating states are possible: (1) the system has the property of quasi-coherence, (2) the system has the property of semi-quasi-coherence, and (3) the system does not have the property of quasi-coherence.

For the two-level decomposition of the serial-parallel production system (PS) chosen in this paper, quasi-coherence occurs in two sets—at the tactical level (Figure 1) and at the operational level (Figure 2). In the case of the series-parallel structure (Figure 1), there are two types of quasi-coherence at the operational level—stream and multi-stream. Hence, quasi-coherence for the case under consideration is defined as follows:

1. Formalization of quasi-coherence at the first decomposition level—the level of separated departments: $^{r(d, \hat{d})}\mathcal{R} := \left( \mathcal{R}_{1(d, d_1)}; \mathcal{R}_{2(d_1, d_2)}; \mathcal{R}_{3(d_2, d_3)}; \dots; \mathcal{R}_{k(d_{k-1}, \hat{d})} \right)$, where $d, d_1, \dots, d_{k-1}, \hat{d} \in \{1, 2, \dots, D\}$.

2. Formalization of quasi-coherence at the second level of decomposition—the level of separated machines: $^{r(m_i(d), m_j(\hat{d}))}\mathcal{R} \; \forall \, i, j \in \{1, 2, \dots, M\} \; \forall \, d, \hat{d} \in \{1, 2, \dots, D\}$, where $d, \hat{d}$ can be equal:

   - Multi-stream case: for a given machine numbered $m\left(\hat{d}\right)$ where $m \in \{1, 2, \dots, M\}, \hat{d} \in \{1, 2, \dots, D\}$ $\exists m_1(d_1), \dots, m_N(d_N)$ : $\exists \left\{ ^{r(m_1(d_1), m(\hat{d}))}\mathcal{R}_1, \, ^{r(m_2(d_2), m(\hat{d}))}\mathcal{R}_2, \dots, \, ^{r(m_N(d_N), m(\hat{d}))}\mathcal{R}_N \right\}$, there is quasi-coherence for $m\left(\hat{d}\right)$ with determined $i$ and $j$;

   - Stream case: for a given machine numbered $m\left(\hat{d}\right)$, where $m \in \{1, 2, \dots, M\}, \hat{d} \in \{1, 2, \dots, D\}$ $\exists m_1(d_1) : \exists! \left\{ ^{r(m_1(d_1), m(\hat{d}))}\mathcal{R}_1, \right\}$, where $m_1(d_1)$ is the beginning machine of the chain $^{r(m_1(d_1), m(\hat{d}))}\mathcal{R}_1$, there is quasi-coherence for a particular $m\left(\hat{d}\right)$.

The definition of $k$th survival value was used to determine the impact of occurring disruptions on the continuity of the processed material flow. A formalisation of the definition of $k$th survival value is presented in the paper [66]. The $k$th survival value is a term developed by the research team to define a stochastic criterion for the stability of a production system. Based on the definition of the survival function $\mathcal{S}$ of the random variable $\mathcal{T}$: $\mathcal{S}(\mathcal{T}) := P(\mathcal{T} > \tau) = 1 - F_\mathcal{T}(\tau)$, $\tau \in \mathbb{R}$, the research team developed the term $k$th survival value [66]. By $k$th survival value, a probability is defined as follows:

$$k_{(T_P, T_B)} := P(T_P - T_B \geq k) = P(T_P \geq k + T_B) = 1 - F_{T_P - T_B}(k), \tag{7}$$

where: $k_{(T_P, T_B)}$—$k$th survival value; $T_P$—random variable of the duration of the system having the quasi-coherence property; $T_B$—random variable of the duration of the system in the state of semi-quasi-coherence or the state of complete loss of the quasi-coherence property; $k$—deterministic value for which the probability $k_{(T_P, T_B)}$ is determined; and $F_{T_P-T_B}(k)$—value of the distribution function of the difference of the $T_P - T_B$ random variables at point $k$.

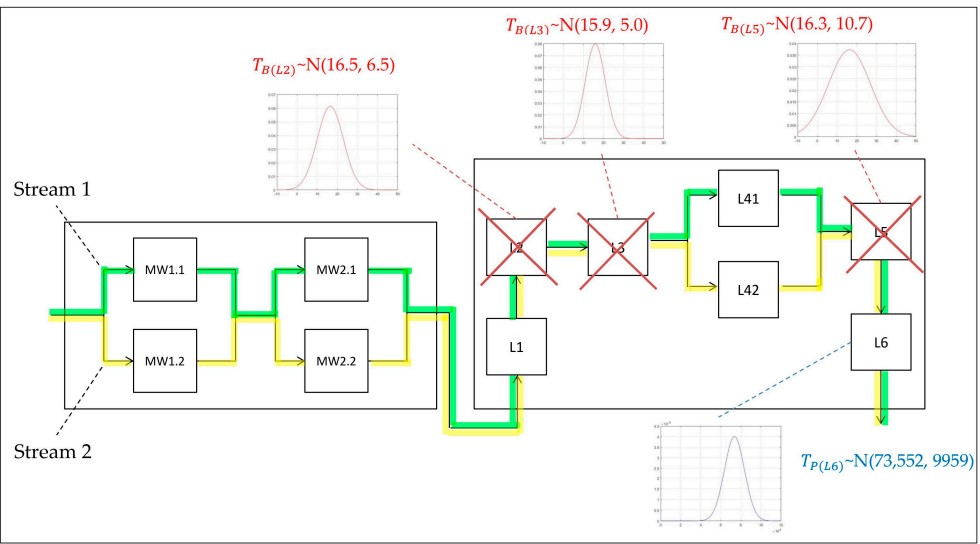

**Figure 2.** Chains of synchronous relations in which a normal distribution is occurring. Two examples of processed material stream are marked in green and yellow.

System survival is understood as the ability of a complex system to correctly perform given functions ($\Phi = \{\Phi_1, \ldots, \Phi_n\}$) under specified operating conditions ($\kappa$) and within a specified time ($\tau \in \mathbb{R}$). The measures of system survival are the probabilities $\mathcal{S}_i(\mathcal{T}) = P_i(\mathcal{T} > \tau)$, $i = 1, \ldots, N$ of performing the tasks specified on the set of $N$ elements of the complex system.

The density function of the difference of two independent random variables with a given distribution is determined from the definition of a convolution function, the formula of which is presented in (8):

$$f_{T_P-T_B}(t) = \begin{cases} \int_t^\infty f_{T_P}(z) \cdot f_{T_B}(z-t)\, dz \,, & for\ t \geq 0 \\ \int\limits_0^\infty f_{T_P}(z) \cdot f_{T_B}(z-t)\, dz \,, & for\ t < 0 \end{cases} \tag{8}$$

where $f_{T_P-T_B}$ is the differential density function of random variables $T_P - T_B$.

Formula (8) shows the case in which the considered arguments of the domain of the random variable $T_P - T_B$ are grouped into values greater than zero ($t \geq 0$) and less than zero ($t < 0$). Depending on the types of distributions of the $T_P$ and $T_B$ random variables, the grouping of the domain of the function $f_{T_P-T_B}(t)$ can have more cases. Knowing the distributions of the density functions of the random variables $T_P$ and $T_B$, we perform the substitution into (8) obtaining the probability distribution function of $f_{T_P-T_B}(t)$, and then we use the $k$th survival value—Formula (7).

On basis of the $k$th survival value, it is possible to analyse the stability of the operation of any technical system, considering their stochastic nature of operation at the same time. In real manufacturing systems, process execution times are variable. This variability is a result of many factors [63,67]. Systems analyses that take the probabilistic nature of systems into account are difficult and time-consuming. The level of difficulty of these analyses increases with the complexity of the system and the number of random variables considered. This article presents the results of the next stage of research related to maintaining the

continuity of the flow of processed materials in a convergent manufacturing system. Hence, the presented algorithm takes into account the factors of variability of production task execution times, variability of the number of elements in a production batch, and variability of the planning route.

### 3. Methodology for the Class of Production Systems Chosen for Consideration

The object chosen for consideration is a series-parallel production structure in which the material flows are convergent. This presented methodology has utilitarian properties and can be applied to any class of production systems. A general schematic of the analysed structure is shown in Figure 1. Detailed analyses of the proposed methodology were made for the systems illustrated in Figures 2–6, in which both linear flows and series-parallel flows occur.

*3.1. Exemplification of the Method When Using Gaussian Distribution for the Random Variables of Duration of Proper Operation $T_P$ and Duration of Breakdown $T_B$*

The Gaussian distribution, also known as the normal distribution, is commonly used to describe the dependence of many random events. Its popularity is due to the properties of the distribution and its easy application. In addition, most cases follow a normal distribution with an assumed acceptable confidence level. Analyses of the performance of real production systems also make use of the properties of the Gaussian distribution. The TPM method used in lean production evaluates machine failure rates according to MTTR, MTTF, and MTBF. These indicators are defined as the arithmetic mean:

-     For MTTR, a set of durations of breakdowns or any type of downtime of a line or individual machine;
-     For MTTF, a set of durations of correct operation between elementary stops;
-     For MTBF, a set of durations between events of individual downtimes.

In order to make the analyses more precise, a detailed classification of the downtimes occurring during the specified time interval is performed and their average values are determined for a separated group. For the accuracy of prevention action planning, it is advisable to determine the spread of the data in relation to the average value. The most commonly used measure of the spread is the standard deviation value. With the mean and standard deviation values, it is possible to make a determination of the percentages of a given variable within $\pm 1\sigma$, $\pm 2\sigma$, and $\pm 3\sigma$ according to a normal distribution. The results of such analyses have been published, for example, in the literature.

In Section 3.1, an analysis of the stability of manufacturing system operation using quasi-coherence and the definition of the $k$th survival value is presented for the case of assuming that the random variables of failure and proper operation follow a normal distribution. Then, the variables $T_P$ and $T_B$ are defined by the mean $\mu_P$ and $\mu_B$, respectively, and the standard deviation $\sigma_P$ and $\sigma_B$, respectively:

$$T_P \sim Normal(\mu_P, \sigma_P) \tag{9}$$

$$T_B \sim Normal(\mu_B, \sigma_B). \tag{10}$$

The use of the Gaussian distribution to approximate and model the duration of correct operation $T_P$ and the duration of breakdown $T_B$ is possible if the histogram plots of the empirical sample are symmetrical and the sample mean is strongly greater than three standard deviations: $\mu_P \gg 3\sigma_P$ and $\mu_B \gg 3\sigma_B$. The requirements highlighted in the previous sentence are fulfilled in the analysed company during the manufacturing of the standardised BOM (bill of materials) components. These are such components of the BOM that are included in the majority of final products in one or more pieces each. For this case, the chain of relations having the quasi-coherence property is defined on two streams, which are illustrated in Figure 2.

Utilising the difference property of random variables defined by a normal distribution with the parameters defined in (9) and (10), a random variable of difference $T_P - T_B$ is obtained, which also follows a Gaussian distribution with the following parameters:

$$T_P - T_B \sim Normal\left(\mu_P - \mu_B, \sqrt{\sigma_P^2 + \sigma_B^2}\right). \tag{11}$$

By substituting the parameters in (11) into the formula for the normal distribution density function, we obtain the formula for the probability function of the difference of the random variables $f_{T_P - T_B}(t)$:

$$f_{T_P - T_B}(t) = \frac{1}{\sqrt{2\pi\left(\sigma_P^2 + \sigma_B^2\right)}} \cdot e^{\left(\frac{-(t-(\mu_P-\mu_B))^2}{2(\sigma_P^2+\sigma_B^2)}\right)}, t \in (-\infty, \infty). \tag{12}$$

The determination of the probability of the $k$th survival value for the difference of the random variables complying with a normal distribution for Parameters (9) and (10) is performed using a standardized normal distribution. The determination of the $k$th survival value defined in (7) is determined as follows:

$$k_{(T_P, T_B)} := P(T_P - T_B \geq k) = P\left(\frac{(T_P - T_B) - \mu_P + \mu_B}{\sqrt{\sigma_P^2 + \sigma_B^2}} \geq \frac{k - \mu_P + \mu_B}{\sqrt{\sigma_P^2 + \sigma_B^2}}\right) = 1 - \Phi\left(\frac{k - \mu_P + \mu_B}{\sqrt{\sigma_P^2 + \sigma_B^2}}\right), \tag{13}$$

where $\Phi\left(\frac{k-\mu_P+\mu_B}{\sqrt{\sigma_P^2+\sigma_B^2}}\right)$ is the value of the distribution of the standard normal distribution at a point $\left(k - \mu_P + \mu_B / \sqrt{\sigma_P^2 + \sigma_B^2}\right)$; the value of the distribution at a given point is taken from the tables.

For the analysed exemplary production system, a subsystem was identified in which Assumptions (9) and (10) are met. Figure 2 shows the separated subsystem, for which the data for calculating the $k$th survival value are included in Table 1. The parameters of the distributions were determined for empirical data from a six-year time period.

**Table 1.** Summary of $T_P$ and $T_B$ duration parameters of the normal distribution.

| Parameters of the Random Variable Breakdown Durations $T_B$ | Parameters of the Random Variable Proper Operation Durations $T_P$ | Value of the Indicator $k$th Survival Value $k_{(T_P, T_B)}$ |
|---|---|---|
| $T_{B(L2)} \sim N(16.5, 6.5)$ <br> $T_{B(L3)} \sim N(15.9, 5.0)$ <br> $T_{B(L5)} \sim N(16.3, 10.7)$ | $T_P \sim N(73,552, 9959)$ | $k_{(T_P, T_{B(L2)})} = 0.3707$ <br> $k_{(T_P, T_{B(L3)})} = 0.2119$ <br> $k_{(T_P, T_{B(L5)})} = 0.0119$ |

The $k$th survival value analysis was performed for a paint shop furnace supply system, of which $MTTF = 73,552$ (min). The mean time of correct operation is defined by a normal distribution with a standard deviation of $\sigma_P = 9959$, and then $T_P \sim N(73,552, 9959)$. The impact of machine failures in the paint shop furnace supply streams was assessed for one type of failure, which occurred on three different machines during the analysis period of time. The selected type of failure does not pose a high risk of affecting production continuity, as it occurs sporadically (eight times on average). In addition, an operator with several months of experience is competent to repair this type of failure unassisted. The values in Table 1, correct operation and failure durations, were determined on the basis of empirical data over a six-year period of time.

Due to the comparable $MTTR \approx 16$ (min) values, for a better illustration of the $k$th survival value probability, the calculations were performed for different values of deterministic time $k$, where one day is two shifts:

$$k_{(T_P, T_{B(L2)})} = P\left(T_P - T_{B(L2)} \geq 76,800\right) \cong 37\% \text{ value for 80 days,}$$

$$k_{(T_P, T_{B(L3)})} = P\left(T_P - T_{B(L3)} \geq 81,600\right) \cong 22\% \text{ value for 85 days,}$$

$$k_{(T_P, T_{B(L5)})} = P\left(T_P - T_{B(L5)} \geq 96,000\right) \cong 1.2 \text{ value for 100 days.}$$

*3.2. Exemplification of the Method When Using the Family of Gamma Distributions for the Random Variables of Duration of Proper Operation $T_P$ and Duration of Breakdown $T_B$*

In many real-world situations, the conditions for using the Gaussian distribution to interpolate empirical data are not met. These are cases where there is positive or negative kurtosis and the histograms of the data distribution are right- or left-skewed, i.e., there is no symmetry with respect to the arithmetic mean. In addition, in the analysis of process execution times and failure durations, the condition of non-negative values of function arguments with more than 99% probability should be met. In a normal distribution, when the mean value of the data set is not strongly greater than three standard deviations, then a significant probability value is placed on the negative arguments of the function. This provides a possibility of approximating negative correct operation durations or breakdown durations, which are impossible to occur in reality. In such cases, it is desirable to use distributions in which the domain of the function is defined on the set of positive real numbers ($\mathbb{R}_+$). Hence, in Section 3.2, the stability of the operation of the production system is determined using quasi-coherence and the definition of the $k$th survival value for random variables formulated by a probability distribution belonging to the Gamma family of distributions. The Gamma distribution is used to define the independent variable of the amount of time since the $n$th event in a Poisson process. Gamma distributions are a family of continuous probability distributions with a carrier defined on the range $\langle 0, \infty \rangle$. These distributions are defined by two parameters: shape—$\alpha$ ($k$ or $l$ designations are also used, where $k, l = \alpha$); and scale—$\beta$ ($\theta$ designation is also used, where $\theta = \frac{1}{\beta}$). The Gamma distribution parameters are $\alpha, \beta \in \mathbb{R}_+$. Depending on the value of the shape parameter, it is possible to distinguish cases of three types of distributions:

- Exponential distribution for $\alpha = 1$ and $\lambda > 0$;
- Erlang distribution for $l \in \mathbb{N}_+ / \{1\}$ and $\lambda > 0$;
- Gamma distribution for $\alpha \in \mathbb{R}_+ / \mathbb{N}_+$ and $\beta > 0$.

The exponential distribution and the Erlang distribution are special cases of the Gamma family of distributions. Sections 3.2.1–3.2.3 consider three variants in determining the stability of system operation according to the $k$th survival value.

3.2.1. The Case of Using an Exponential Distribution for the Random Variables of Duration of Proper Operation $T_P$ and Duration of Breakdown $T_B$

The use of an exponential distribution to determine the random variables [68,69] of the duration of correct operation $T_P$ and the duration of any type of breakdown $T_B$ is possible if $T_P$ and $T_B$ are independent, stationary, and memoryless. These conditions for $T_P$ and $T_B$ are fulfilled for the system illustrated in Figure 3. Then,

$$T_P \sim Exp(\lambda_P) \tag{14}$$

$$T_B \sim Exp(\lambda_B), \tag{15}$$

where $\lambda_P$, $\lambda_B$ are the scale parameters of the random variables, respectively, $T_P$ and $T_B$.

Applying the law of total probability for the difference of the random variables $T_P - T_B$ consistent with (13) and (14), the density function $f_{T_P - T_B}(t)$ has the following form:

$$f_{T_P - T_B}(t) = f_{(T_P - T_B)|(T_P - T_B)<0}(t) \cdot P(T_P - T_B < 0) + f_{(T_P - T_B)|(T_P - T_B)\geq 0}(t) \cdot P(T_P - T_B \geq 0). \tag{16}$$

On the basis of the function convolution (8), and after performing transformations, the following is obtained:

$$f_{T_P - T_B}(t) = \begin{cases} \frac{\lambda_P \lambda_B}{\lambda_P + \lambda_B} \cdot e^{-\lambda_P t}, & dla\ t \geq 0 \\ \frac{\lambda_P \lambda_B}{\lambda_P + \lambda_B} \cdot e^{\lambda_B t}, & dla\ t < 0 \end{cases}. \tag{17}$$

Stability analyses of production system operation refer to values of $t \geq 0$; hence, the $k$th survival value of the difference of two random variables following an exponential distribution is determined according to the formula:

$$k_{(T_P, T_B)} = P(T_P - T_B \geq k) = \int_k^\infty f_{T_P - T_B}(t)dt = \frac{\lambda_P \lambda_B}{\lambda_P + \lambda_B} \int_k^\infty \left( e^{-\lambda_P t} \right) dt \tag{18}$$

where $\lambda_P$, $\lambda_B$ are the shape parameters of the random variables $T_P$, $T_B$, respectively, determined according to the formulas:

$$\lambda_P = \frac{n_P}{\sum_{i=1}^n x_{P(i)}} \tag{19}$$

$$\lambda_B = \frac{n_B}{\sum_{i=1}^n x_{B(i)}}, \tag{20}$$

where $x_{P(i)}$, $x_{B(i)}$ are the empirical values of random events interpolated by an exponential distribution for the following sets: durations of correct operation labelled P and durations of breakdown labelled B; $n_P$, $n_B$ are the sample sizes for which the shape parameter of the exponential distribution is determined.

In order to calculate the $k$th survival probability value, the subsystem fulfilling conditions (14) and (15) was extracted. Table 2 shows the empirical data of the example under consideration.

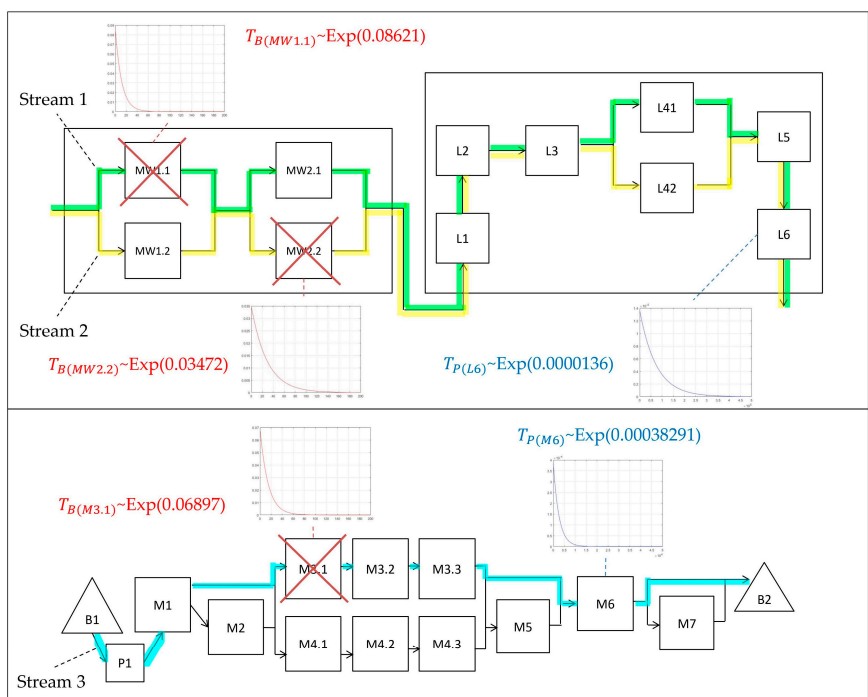

**Figure 3.** Chains of synchronous relations in which there are exponential distributions. Three examples of processed material streams are marked in green, yellow and blue.

The analyses of the difference of the correct operation time random variable $T_P$ and the failure duration random variable $T_B$ are defined by the exponential distribution—Formulas (14) and (15), which refer to the determination of the probability value—and Formula (18), where the correct operation will be longer than the deterministic value of $k$. For the variants shown in Table 2, $k$ takes values equal to the $MTTF$, respectively, for machines $T_{P(L6)} : k = 73,552$ (min) and $T_{P(M6)} : k = 2612$ (min).

**Table 2.** Summary of $T_P$ and $T_B$ time parameters of the exponential distribution.

| Parameters of the Random Variable Breakdown Durations $T_B$ | Parameters of the Random Variable Proper Operation Durations $T_P$ | Value of the Indicator $k$th Survival Value $k_{(T_P, T_B)}$ |
|---|---|---|
| $T_{B(MW1.1)} \sim Exp(0.08621)$ $T_{B(MW2.2)} \sim Exp(0.03472)$ $T_{B(M3.1)} \sim Exp(0.06897)$ | $T_{P(L6)} \sim Exp(0.0000136)$ $T_{P(M6)} \sim Exp(0.00038291)$ | $k_{(T_{P(L6)}, T_{B(MW1.1)})} = 0.36771$ $k_{(T_{P(L6)}, T_{B(MW2.2)})} = 0.36762$ $k_{(T_{P(M6)}, T_{P(M3.1)})} = 0.36579$ |

For the variants considered, $k_{(T_P, T_B)} = P(T_P - T_B \geq k) \approx 36\%$. The interpretation of the results means that at 64%, the operation of machines $L6$ and $M6$ will be interrupted due to failures of objects $MW1.1$, $MW2.2$, and $M3.1$. The value of 64% defines the risk of disruption of the flow of processed material at time $t_{L6} < 73,552$ (min) and $t_{M6} < 2612$ (min). It should be noted that the variant $T_P \sim Exp(\lambda_P)$ minus $T_B \sim Exp(\lambda_B)$ refers to the comparison of individual durations of correct operation and durations of failures. In real cases, a certain number of breakdowns of the same type may occur in a given $\Delta t$, e.g., equal to the $MTTF$. In such cases, the multiplicity of occurrence of the number of breakdowns of the same parameter $\lambda_B$ should be taken into account in the calculations using an Erlang distribution. The following considerations include a longer time scale in which $l_P$ is the number of times of correct operation and $l_B$ is the number of times of failure where the same parameters $\lambda_P$ and $\lambda_B$ occurred.

### 3.2.2. The Case of Using an Erlang Distribution for the Random Variables of Duration of Proper Operation $T_P$ and Duration of Breakdown $T_B$

The Erlang distribution belongs to the family of Gamma distributions for which the shape parameter is a natural number greater than one. It should be noted that for the sum of two independent random variables following an exponential distribution with the same value of the parameter $\lambda$, the resulting Erlang distribution will have parameters $(2, \lambda)$. The general dependence of the exponential distribution $Exp(\lambda)$ on the Erlang distribution can be described as follows:

$$X \sim Erlang(l, \lambda) = \sum_{i=1}^{l} Exp(\lambda) \tag{21}$$

where $l$ is the shape parameter of the Erlang distribution, which also specifies the number of the sum of independent random variables following an exponential distribution, and $\lambda$ is the scale parameter of the exponential and Erlang distributions.

The determination of the stability of the operation of a production system having the quasi-coherence property for chains of synchronous relations using the definition of the $k$th survival value was made for the parameters $\lambda_P \neq \lambda_B$. Then,

$$T_P \sim Erlang(l_P, \lambda_P) \tag{22}$$

$$T_B \sim Erlang(l_B, \lambda_B), \tag{23}$$

where $l_P, l_B \in \mathbb{N}_+ / \{1\}$ and $\lambda_P, \lambda_B \in \mathbb{R}_+$.

Using the definition of the convolution of functions—Formula (8)—and after transformations, the following is obtained:

$$f_{T_P - T_B}(t) = \begin{cases} \frac{\lambda_P{}^{l_P} \cdot \lambda_B{}^{l_B}}{(l_P-1)! \cdot (l_B-1)!} \cdot e^{\lambda_B \cdot t} \int\limits_{t}^{\infty} z^{l_P-1}(z-t)^{l_B-1} \cdot e^{-(\lambda_P+\lambda_B) \cdot z} \, dz \,, & for \ t \geq 0 \\ \frac{\lambda_P{}^{l_P} \cdot \lambda_B{}^{l_B}}{(l_P-1)! \cdot (l_B-1)!} \cdot e^{-\lambda_P \cdot t} \int\limits_{-t}^{\infty} z^{l_B-1}(z+t)^{l_P-1} \cdot e^{-(\lambda_P+\lambda_B) \cdot z} \, dz \,, & for \ t < 0 \end{cases} \tag{24}$$

Failure analyses of complex production systems refer to cases where $t \geq 0$; hence, the $k$th survival value is determined as follows:

$$k_{(T_P,T_B)} = P(T_P - T_B \geq k) = \int\limits_{k}^{\infty} \frac{\lambda_P{}^{l_P} \cdot \lambda_B{}^{l_B}}{(l_P-1)! \cdot (l_B-1)!} \cdot e^{\lambda_B \cdot t} \left( \int\limits_{t}^{\infty} z^{l_P-1}(z-t)^{l_B-1} \cdot e^{-(\lambda_P+\lambda_B) \cdot z} \, dz \right) dt \tag{25}$$

where $l_P, l_B, \lambda_P, \lambda_B$ are the shape ($l$) and scale ($\lambda$) parameters of the random variables $T_P$ and $T_B$, respectively.

The situation defined by Formulas (22) and (23) referring to the case when there is $T_P \sim Erlang(l_P, \lambda_P)$ and $T_B \sim Erlang(l_B, \lambda_B)$, for $\lambda_P \neq \lambda_B$ and $l_P, l_B > 1$ on a period $\Delta t$, is illustrated in Figure 4. In the analysed $\Delta t$, the machine L6 executes the processing of $l_P$ components, while at the same time $\Delta t$, there are $l_B$ downtimes of the same type, e.g., failures originating from the same distribution. This situation can happen when the type of downtimes under consideration (e.g., failures) occurs on one machine or on several different machines. These analyses can be performed for the loss of the quasi-coherence property in one chain of relations or the whole set of chains of relations supplying the considered machine L6. The empirical data for calculating the described alternative are summarized in Table 3.

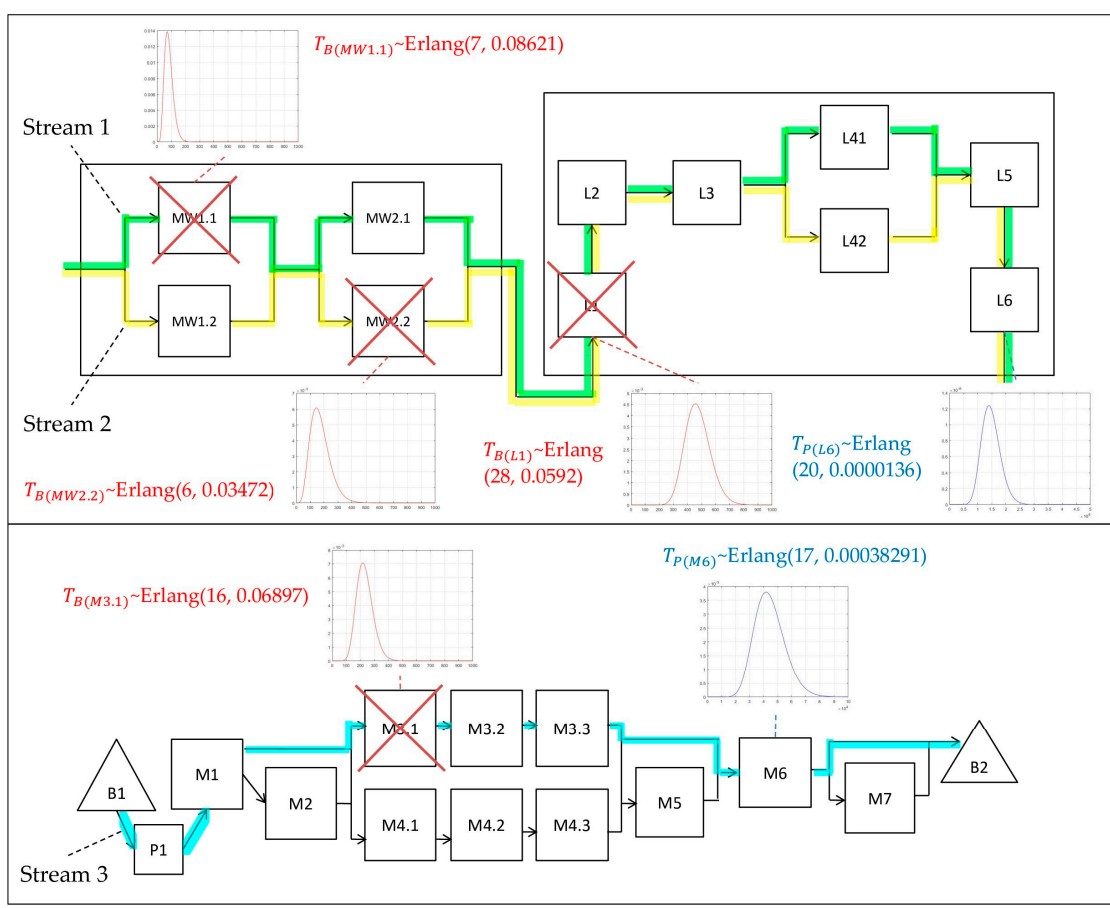

**Figure 4.** Chains of synchronous relations in which Erlang distributions occur. Three examples of processed material streams are marked in green, yellow and blue.

An exemplification of the quasi-coherence and $k$th survival value algorithm using Erlang distributions for $T_P$ correct operation durations and $T_B$ failure durations was performed for two separate systems: (1) relations supplying the paint shop and (2) relations supplying the glass tempering line. These two systems are characterised by a different flow structure, as illustrated in Figure 1. The paint shop supply relations subsystem is a series-parallel system with multiple independent relations chains with different labour efficiencies. This variability shows a dynamic variation in the levels of labour intensity of works-in-progress depending on the timeframe adopted. The glass tempering line system is serial, in which the value of the level of labour at each station (machine) is stationary, i.e., its dynamics of variation for an assumed $\Delta t$ is constant. In a series system, the property of semi-quasi-coherence cannot occur because the failure of any machine belonging to the chain of relations determines the loss of quasi-coherence. Furthermore, in the production system chosen for consideration, the glass tempering line is a new production unit, the actual capacity of which varies between 40% and 60% of the nominal capacity.

The following variants were chosen for the analyses:

1. The supply subsystem of the automatic paint shop line:
   - The highest energy demand in the line is for the furnace; hence, the reference to the correct operation duration of this module is $MTTF\left(T_{P(L6)}\right) = 73,552$ (min);
   - One type of failure occurring on three different machines belonging to three different relation chains was assumed, where for each object at a given $\Delta t$ the following was calculated: $MTTR\left(T_{B(L1)}\right) = 16.9$ (min); $MTTR\left(T_{B(MW1.1)}\right) = 11.6$ (min); $MTTR\left(T_{B(MW2.2)}\right) = 28.8$ (min).

2. The subsystem of the glass tempering line:
   - The mean time of the correct operation of the glass tempering furnace is $MTTF\left(T_{P(M6)}\right) = 2611.6$ (min);
   - The mean time of one type of failure of a machine belonging to the chain of relations supplying the furnace is $MTTR\left(T_{B(M3.1)}\right) = 14.5$ (min).

The parameters of the Erlang distributions for the objects considered are shown in Table 3.

**Table 3.** Summary of the $T_P$ and $T_B$ duration parameters of the Erlang distribution.

| Parameters of the Random Variable Breakdown Durations $T_B$ | Parameters of the Random Variable Proper Operation Durations $T_P$ | Value of the Indicator $k$th Survival Value $k_{(T_P, T_B)}$ |
|---|---|---|
| $T_{B(L1)} \sim Erlang(28, 0.05920)$ $T_{B(MW1.1)} \sim Erlang(7, 0.08621)$ $T_{B(MW2.2)} \sim Erlang(6, 0.03472)$ $T_{B(M3.1)} \sim Erlang(16, 0.06897)$ | $T_{P(L6)} \sim Erlang(20, 0.00001360)$ $T_{P(M6)} \sim Erlang(17, 0.00038291)$ | $k_{(T_{P(L6)}, T_{B(L1)})} = 0.46914$ $k_{(T_{P(L6)}, T_{B(MW1.1)})} = 0.46961$ $k_{(T_{P(L6)}, T_{B(MW2.2)})} = 0.46950$ $k_{(T_{P(M6)}, T_{P(M3.1)})} = 0.55690^*$ |

In Table 3, the $k$th survival value of the glass tempering line system is determined for 60% utilisation of the nominal line capacity—$k_{(T_{P(M6)}, T_{P(M3.1)})} = 0.5569^*$. Table 4 shows the $k$th survival value calculations for 40%, 60%, and 80% nominal capacity utilisation levels.

**Table 4.** Summary of $T_P$ and $T_B$ durations of Erlang distribution and different levels of line utilisation.

| Parameters of the Random Variable Breakdown Durations $T_B$ | Parameters of the Random Variable Proper Operation Durations $T_P$ | Value of the Indicator $k$th Survival Value $k_{(T_P, T_B)}$ |
|---|---|---|
| $T_{B(M3.1)} \sim Erlang(16, 0.06897)$ $T_{B(M3.1)} \sim Erlang(16, 0.06897)$ $T_{B(M3.1)} \sim Erlang(16, 0.06897)$ | $T_{P(M6)} \sim Erlang(17, 0.00038291)$ | $k_{(T_{P(M6)}, T_{P(M3.1)})}^{40\%} = 0.998899$ $k_{(T_{P(M6)}, T_{P(M3.1)})}^{60\%} = 0.556898$ $k_{(T_{P(M6)}, T_{P(M3.1)})}^{80\%} = 0.036338$ |

For the variant analysed above, there may be a situation in a real production system where, at a given $\Delta t$, there is only one event of loss of the quasi-coherence property. Then, $T_P \sim Erlang(l_P, \lambda_P)$, while for either a single relations chain or a set of relations chains supplying the machine M6, $T_B \sim Erlang(1, \lambda_B)$, i.e., $T_B \sim Exp(\lambda_B)$. This variant is shown in Figure 5. Then, the density function $f_{T_P - T_B}$ of the difference of the random variables $T_P - T_B$ after transformations and substitutions is defined as follows:

$$f_{T_P-T_B}(t) = \begin{cases} \frac{\lambda_B \cdot \lambda_P^{l_P}}{(l_P-1)!} \cdot e^{\lambda_B \cdot t} \int\limits_t^\infty z^{l_P-1}(z-t)^{l_B-1} \cdot e^{-(\lambda_P+\lambda_B)\cdot z} \, dz, & for \ t \geq 0 \\ \frac{\lambda_B \cdot \lambda_P^{l_P}}{(l_P-1)!} \cdot e^{-\lambda_P \cdot t} \int\limits_{-t}^\infty (z+t)^{l_P-1} \cdot e^{-(\lambda_P+\lambda_B)\cdot z} \, dz, & for \ t < 0 \end{cases}. \tag{26}$$

As in the previous cases, the analyses of the non-continuity of the flow of processed materials refer to $t \geq 0$; hence,

$$k_{(T_P, T_B)} = P(T_P - T_B \geq k) = \int\limits_k^\infty f_{T_P-T_B}(t)dt = \int\limits_k^\infty \frac{\lambda_B \cdot \lambda_P^{l_P}}{(l_P-1)!} \cdot e^{\lambda_B \cdot t} \left( \int\limits_t^\infty z^{l_P-1}(z-t)^{l_B-1} \cdot e^{-(\lambda_P+\lambda_B)\cdot z} \, dz \right) dt \tag{27}$$

For the variant formalised by Formula (27), it was assumed that in the specified timeframe $\Delta t$, one failure could occur with a mean time to repair $MTTR\left(T_{B(MW2.2)}\right) = 28.8$ (min) or $MTTR\left(T_{B(MW1.1)}\right) = 11.6$ (min). The analysis was performed for the supply relations chain subsystem of the automatic paint shop line. Figure 5 illustrates the variants of the $k$th survival value analyses shown in Tables 5 and 6.

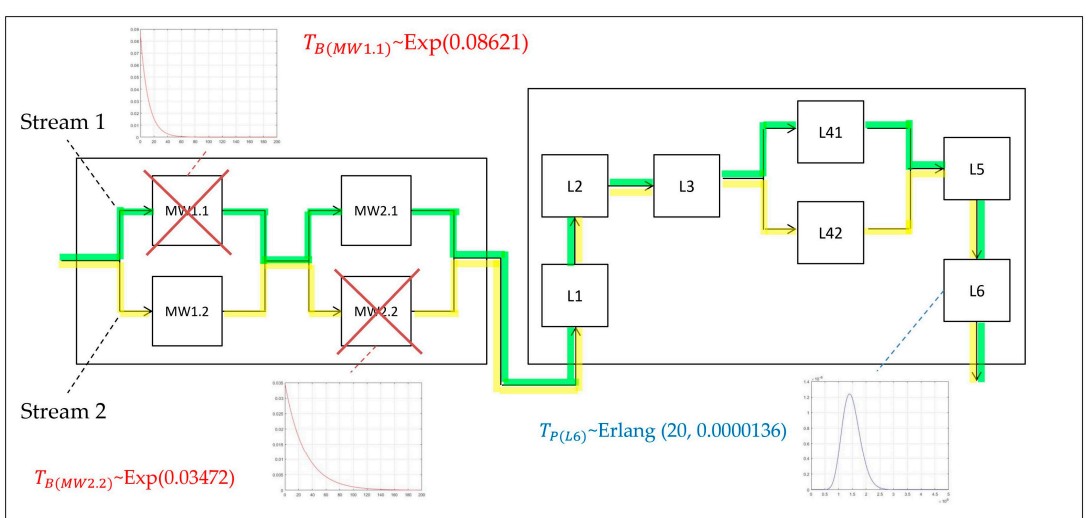

**Figure 5.** Synchronous relation chains in which there are exponential and Erlang distributions. Two examples of processed material streams are marked in green and yellow.

Tables 5 and 6 summarise the three cases, which, respectively, refer to different levels of $k$(min) values. $MTTF\left(T_{P(L6)}\right) = 73,552$(min) and $MTTR\left(T_{B(MW2.2)}\right) = 28.8$(min)—Table 5; $MTTR\left(T_{B(MW1.1)}\right) = 11.6$(min)—Table 6.

-    I—$k_{(T_P, T_B)} = P(T_P - T_B \geq 1,471,040)$;
-    II—$k_{(T_P, T_B)} = P(T_P - T_B \geq 1,324,017)$;
-    III—$k_{(T_P, T_B)} = P(T_P - T_B \geq 1,176,913)$.

**Table 5.** Summary of the parameters of the durations $T_P \sim Erlang$ and $T_B \sim Exp(0.0347)$.

| Parameters of the Random Variable Breakdown Durations $T_B$ | Parameters of the Random Variable Proper Operation Durations $T_P$ | Value of the Indicator $k$th Survival Value $k_{(T_P,T_B)}$ |
|---|---|---|
| $T_{B(MW2.2)} \sim Exp(0.03472)$ | | $k_{(T_P,T_{B(MW2.2)})}^{I} = 0.469677$ |
| $T_{B(MW2.2)} \sim Exp(0.03472)$ | $T_P \sim Erlang(20, 0.0000136)$ | $k_{(T_P,T_{B(MW2.2)})}^{II} = 0.650293$ |
| $T_{B(MW2.2)} \sim Exp(0.03472)$ | | $k_{(T_P,T_{B(MW2.2)})}^{III} = 0.811713$ |

**Table 6.** Summary of the parameters of the durations $T_P \sim Erlang$ and $T_B \sim Exp(0.08621)$.

| Parameters of the Random Variable Breakdown Durations $T_B$ | Parameters of the Random Variable Proper Operation Durations $T_P$ | Value of the Indicator $k$th Survival Value $k_{(T_P,T_B)}$ |
|---|---|---|
| $T_{B(MW1.1)} \sim Exp(0.08621)$ | | $k_{(T_P,T_{B(MW1.1)})}^{I} = 0.469610$ |
| $T_{B(MW1.1)} \sim Exp(0.08621)$ | $T_P \sim Erlang(20, 0.0000136)$ | $k_{(T_P,T_{B(MW1.1)})}^{II} = 0.650314$ |
| $T_{B(MW1.1)} \sim Exp(0.08621)$ | | $k_{(T_P,T_{B(MW1.1)})}^{III} = 0.811816$ |

The $k$th survival values in Tables 5 and 6 refer to different timeframes: $k^I > k^{II} > k^{III}$. These results should be interpreted as follows. For a decreasing value of $k$, the probability of failure after time $P(T_P - T_B \geq k)$ increases. Hence, correspondingly, the risk of breakdown of the line at time $\Delta t < k$ decreases.

3.2.3. Gamma Distribution Use Case for $T_P$ and $T_B$

The estimation of random variables by the Gamma distribution applies to events in which the function domain is defined on the set of positive real numbers ($\mathbb{R}_+$). The Gamma distribution formalises the general form of the predefined cases of the exponential distribution and the Erlang distribution, as it takes into account non-integer values of the shape parameter ($\alpha$). If the random variables $T_P$ and $T_B$ are defined as

$$T_P \sim Gamma(\alpha_P, \beta_P) \tag{28}$$

$$T_B \sim Gamma(\alpha_B, \beta_B). \tag{29}$$

Then, using the law of total probability for the difference of the $T_P - T_B$ random variables, the probability density function $f_{T_P - T_B}$ has the form:

$$f_{T_P-T_B}(t) = \begin{cases} \frac{\beta_P{}^{\alpha_P} \cdot \beta_B{}^{\alpha_B}}{\Gamma(\alpha_P) \cdot \Gamma(\alpha_B)} \cdot e^{\beta_B \cdot t} \int\limits_{t}^{\infty} z^{\alpha_P - 1}(z - t)^{\alpha_B - 1} \cdot e^{-(\beta_P + \beta_B) \cdot z}\, dz\,, & for \ t \geq 0 \\ \frac{\beta_P{}^{\alpha_P} \cdot \beta_B{}^{\alpha_B}}{\Gamma(\alpha_P) \cdot \Gamma(\alpha_B)} \cdot e^{-\beta_P \cdot t} \int\limits_{-t}^{\infty} z^{\alpha_B - 1}(z + t)^{\alpha_P - 1} \cdot e^{-(\beta_P + \beta_B) \cdot z}\, dz\,, & for \ t < 0 \end{cases} \tag{30}$$

where $\Gamma(\alpha_P)$, $\Gamma(\alpha_B)$ is the value of the Gamma function at the points, respectively, $\alpha_P$ and $\alpha_B$.

After the transformations and appropriate substitutions, the $k$th survival value for $k \geq 0$ is calculated according to the formula:

$$k_{(T_P,T_B)} = \int\limits_{k}^{\infty} f_{T_P-T_B}(t)dt = \int\limits_{k}^{\infty} \frac{\beta_P{}^{\alpha_P} \cdot \beta_B{}^{\alpha_B}}{\Gamma(\alpha_P) \cdot \Gamma(\alpha_B)} \cdot e^{\beta_B \cdot t} \left( \int\limits_{t}^{\infty} z^{\alpha_P - 1}(z - t)^{\alpha_B - 1} \cdot e^{-(\beta_P + \beta_B) \cdot z}\, dz \right) dt \tag{31}$$

An example of the use of the Gamma distribution to estimate the duration of correct operation and breakdowns in a $k$th survival value model is shown below.

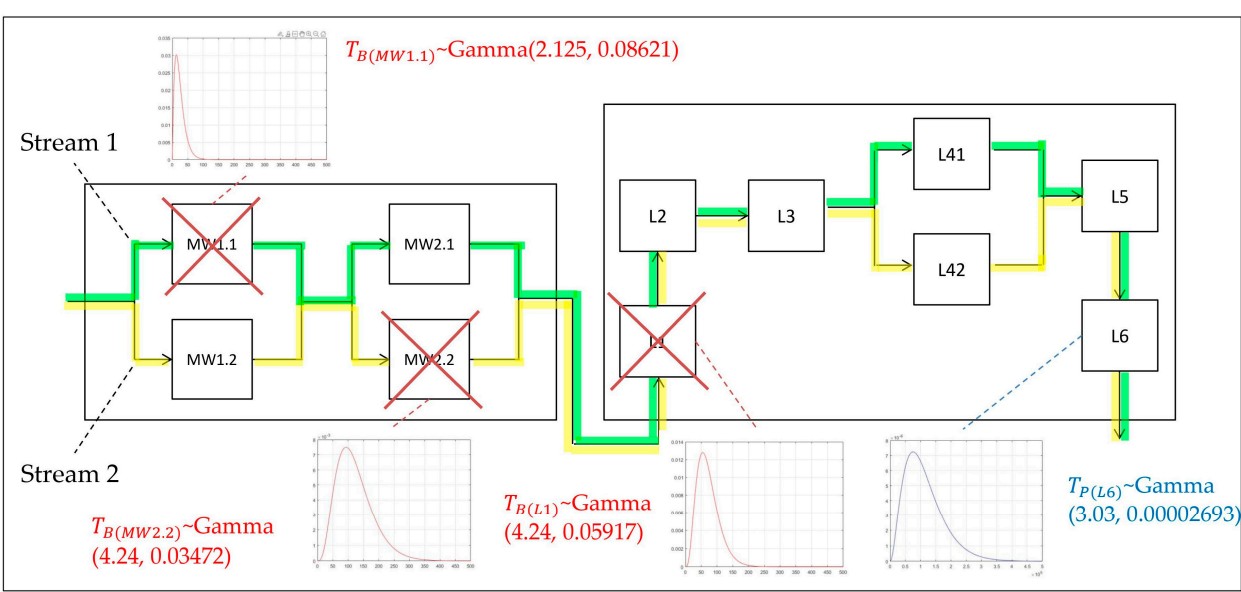

**Figure 6.** Chains of synchronous relations in which Gamma distributions occur. Two examples of processed material streams are marked in green and yellow.

The Gamma distribution is a generalization of the exponential and Erlang distributions. The failure analyses of the gathered empirical data include the $\alpha \in \mathbb{N}_+$ cases. Selecting a different type and kind of failure is still the case of subtracting exponential or Erlang distributions or a combination thereof will still be considered. In the *k*th survival value probability validation step of random variables following the Gamma distribution, the MTTF and MTTR values were averaged over the timeframe $\Delta t = 240,000$ (min)—a value corresponding to 250 days/year in a two-shift operation. The shape parameter of the Gamma distributions is then assumed to have non-integer values: $\alpha \in \mathbb{R}_+/\mathbb{N}_+$. The analysis performed for the data is shown in Table 7.

**Table 7.** Summary of the parameters of the $T_P$ and $T_B$ durations of the Gamma distribution.

| Parameters of the Random Variable Breakdown Durations $T_B$ | Parameters of the Random Variable Proper Operation Durations $T_P$ | Value of the Indicator *k*th Survival Value $k_{(T_P,T_B)}$ |
|---|---|---|
| $T_{B(MW1.1)} \sim Gamma(2.125, 0.08621)$ $T_{B(L1)} \sim Gamma(4.24, 0.059172)$ $T_{B(MW2.2)} \sim Gamma \cdot (4.24, 0.034722)$ | $T_P \sim Gamma(3.03, 2.6936 \cdot 10^{-5})$ | $k_{(T_P,T_{B(MW1.1)})} = 0.688462$ $k_{(T_P,T_{B(L1)})} = 0.688121$ $k_{(T_P,T_{B(MW2.2)})} = 0.687755$ |

The table summarises the *k*th survival values for the rescaled data in relation to operation timeframe $\Delta t = 240,000$ (min). The data for which prior calculations were made cover a period more than six times longer. The *k*th survival value showed an approximately 20% increase in relation to the base period. This is due to the proportional reduction in the number of events at a given $\Delta t$ and is directly influenced by the value of the standard deviation at that time.

## 4. Results and Discussion of Future Studies

This paper presents a model for determining the probability of the on-time realisation of a set of production tasks, taking into account the values of failure indicators of machines. Chains of convergent material flow relations were the objects of consideration. The main assumption of the developed model was to reduce (or completely eliminate) the idle operation of machines with the highest energy demand.

Sections 3.2.1–3.2.3 present a formalisation of the *k*th survival value model for the cases of estimation of durations $T_P$ and $T_B$ using distributions from the Gaussian and

Gamma family. In this presented model, variants were considered for the probability of the difference of the set of durations of correct operation on the technical object with the highest energy demand in relation to the set of durations of failures on the object belonging to the chain of relation supplying this machine. The calculations were performed for the case $T_P - T_B$, where the respective variables, both $T_P$ and $T_B$, originate from the same distribution. The case $T_P - T_B$ was not considered where $T_B = \sum_{i=1}^{n} \sum_{j=1}^{k_i} Gamma(\alpha_{i,k_i}, \beta_{i,k_i})$.

In the next step of this study, the case will be complemented where, in the set of synchronous relations supplying the machine with the highest energy demand, several different downtimes will occur with different distributions belonging to the Gamma family of distributions, i.e., a variant will be considered where, for a given $\Delta t$ on the $i$th machine, the $j$th failure will occur where $i = 1, 2, \ldots, n$ and $j = 1, 2, \ldots, k_i$. Then, the probability density function of the occurrence of a downtime on any technical object is defined by the Formula in [70]:

$$f_{T_B}(t) = \sum_{i=1}^{n} \lambda_i{}^{k_i} \cdot e^{-t \cdot \lambda_i} \sum_{j=1}^{k_i} \frac{(-1)^{k_i-j}}{(j-1)!} \cdot t^{j-1} \sum_{\substack{m_1 + \ldots + m_n = k_i - j \\ m_i = 0}} \prod_{\substack{l=1 \\ l \neq i}}^{n} \binom{k_l + m_l - 1}{m_l} \frac{\lambda_l{}^{k_l}}{(\lambda_l - \lambda_i)^{k_l + m_l}} \tag{32}$$

If, for a sufficiently long timeframe for the object under consideration, there are production processes of multiple batches numbered $l = 1, 2, \ldots, n_p$, and in each batch, there is a different number of elements to be manufactured ($h = 1, 2, \ldots, H_l$), then the time $T_P$ is determined by the function:

$$f_{T_P}(t) = \sum_{l=1}^{n_p} \lambda_l{}^{k_l} \cdot e^{-t \cdot \lambda_l} \sum_{h=1}^{H_l} \frac{(-1)^{H_l-j}}{(j-1)!} \cdot t^{h-1} \sum_{\substack{m_1 + \ldots + m_{n_p} = H_l - h \\ m_l = 0}} \prod_{\substack{s=1 \\ s \neq l}}^{n_p} \binom{k_s + m_s - 1}{m_s} \frac{\lambda_s{}^{h_s}}{(\lambda_s - \lambda_l)^{h_s + m_s}} \tag{33}$$

In order to calculate the $k$th survival value for the difference of independent random variables, the functions defined by Formulas (32) and (33) should be convolved. This variant will take into account all possible cases of flexible production, with the number of elements to be manufactured varying in time, with different types of failures occurring with a different frequency on different technical objects.

In the article presented, the $k$th survival value algorithm takes into account the randomness of the operating and failure times. Gaussian and Gamma distributions were included in the considerations. The validation showed a high sensitivity of the model to the values of the distribution parameters. According to the authors, this will result in the accuracy of the calculations and their possible inaccuracy for the assumed approximations of the empirical data. During the calculations, it was noted that there was a close relation between the parameters of the distributions—$(l_P, \lambda_P \Longleftrightarrow l_B, \lambda_B)$ and $(\alpha_P, \beta_P \Longleftrightarrow \alpha_B, \alpha_B)$. The cases presented in this paper did not show whether this relation is interactional or correlational. To determine this, a much larger series of trials should be carried out for different cases.

In the step of calculating the $k$th survival values, the possibility of the existence of a limit to the applicability of the distribution $f_{T_P - T_B}(t)$, with close interdependence of the parameters $(l_P, \lambda_P \Longleftrightarrow l_B, \lambda_B)$ and $(\alpha_P, \beta_P \Longleftrightarrow \alpha_B, \alpha_B)$, as well as the parameter $k$, for which $P(T_P - T_B \geq k)$ is determined, was also recognised. In order to confirm this thesis, it is necessary to develop an application that allows the graphical representation of $f_{T_P - T_B}(t)$ for an arbitrary value of $k \in \mathbb{R}_+$, $T_P \sim Gamma(\alpha_P, \beta_P)$, and $T_B \sim Gamma(\alpha_B, \beta_B)$. It is also crucial to confirm the assumptions made with the necessary calculations. Development work is currently in progress in this area.

## 5. Conclusions

Modelling the operational reliability of real production systems is a complex task. The level of difficulty increases as the number of parameter variables considered in the model increases. This paper presents the author's model for determining the $k$th survival value.

This proposed algorithm is used to determine the probability value of the difference of random variables of correct operation duration and failure duration with given probability density functions.

This paper formalises a complex manufacturing system according to the general theory of complex systems [71,72]. A two-level decomposition of the manufacturing system was made, separating a subset of elements of the departmental subsystem and a subset of elements of the operational level (manufacturing machine level). Then, the property of quasi-coherence and semi-quasi-coherence on sets of synchronous relations was defined. Furthermore, the cases of relations chains where neither the quasi-coherence nor semi-quasi-coherence property occurs were defined.

The methodology developed is intended to determine the risk of potential line downtimes in production systems containing machines with the highest energy demand. Hence, for separated chains of relations containing production processes with the use of an automatic paint shop furnace and a glass tempering furnace, the impact of failures on objects supplying high-energy demand machines was analysed. The validation of the presented model was performed for independent random variables defined by distributions:

- $T_P \sim Normal(\mu_P, \sigma_P)$ and $T_B \sim Normal(\mu_B, \sigma_B)$, and then the random variable of difference $T_P - T_B \sim Normal\left(\mu_P - \mu_B, \sqrt{\sigma_P^2 + \sigma_B^2}\right)$;
- $T_P \sim Exp(\lambda_P)$ and $T_B \sim Exp(\lambda_B)$;
- $T_P \sim Erlang(l_P, \lambda_P)$ and $T_B \sim Erlang(l_B, \lambda_B)$;
- $T_P \sim Erlang(l_P, \lambda_P)$ and $T_B \sim Exp(\lambda_B)$;
- $T_P \sim Gamma(\alpha_P, \beta_P)$ and $T_B \sim Gamma(\alpha_B, \beta_B)$.

For the family of Gamma distributions, the probability density functions $f_{T_P-T_B}(t)$ were determined using the definition of a convolution of functions. Various variants of the calculation of the *k*th survival value are summarized in Tables 1–7. In real production systems, an increased confidence interval is accepted at the expense of a reduction in the time effort resulting from a more accurate estimation of the fit. Hence, simplifications and generalisations are made to allow the use of Gaussian distributions.

**Author Contributions:** Conceptualisation, B.Z.; methodology, B.Z.; software (Matlab R2022a), J.W.; validation, B.Z. and J.W.; formal analysis, B.Z.; investigation, B.Z.; resources, J.W. and B.Z.; data curation, J.W. and B.Z.; writing—original draft preparation, B.Z.; writing—review and editing, J.W.; visualisation, J.W.; supervision, B.Z.; project administration, B.Z.; funding acquisition, B.Z. and J.W. All authors have read and agreed to the published version of the manuscript.

**Funding:** This research was funded by AGH University of Krakow, grant number 16.16.130.943/ ZWOLIŃSKA and "The APC was funded by Grant Dziekana 16.16.130.943/ZWOLIŃSKA".

**Data Availability Statement:** The data presented in this study are available upon request from the corresponding author. The data are not publicly available due to the fact that the data are the property of the authors.

**Conflicts of Interest:** The authors declare no conflict of interest. The funders had no role in the design of this study, in the collection, analyses, or interpretation of data, in the writing of the manuscript, or in the decision to publish the results.

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
