# Peer review of "Modelling the Reliability of Logistics Flows in a Complex Production System"

_energies, doi:10.3390/en16248071_

Round 1
Reviewer 1 Report
Comments and Suggestions for Authors
The authors present a methodology to evaluate the impact of a machine's downtime on the production system. The organization of the manuscript, however, does not explicit the outcomes of the work.
This is because the section "Materials and Methods" includes results that should be placed in another section and this way makes the contributions of the work more clear.
Therefore, I would recommend conveniently splitting section 2 into one or more sections and even merging these new sections with the original sections "Results" and "Discussion".
Also, the section "Results" has some content, e.g., the three areas in which the proposed model can be applied (line 554 to 565), that could integrate the section "Materials and Methods".
In fact, from the sections "Results" and "Discussion" it seems that something is missing in the work, I mean, the objectives were not met. Thus, I would suggest reorganizing the manuscript's structure (as mentioned above) and resetting the aim of the paper.
In addition:
Please revise the abbreviations: SME was not defined in line 43; PS in line 91 should be placed between parentheses; TPM, MTTR, MTTF, MTBF were defined twice (in lines 278-281).
Please correct the following mistakes:
. in line 105 - 'presented' in place of "resented"
. in line 270 - 'number' in place of "n umber"
. in line 297 - 'and' in place of "oraz"
. in line 306 - 'in figure 2' in place of "in the figure 2"
. in line 326 - 'performed' in place of "performer"
. in line 330 - 'one type of failure' in place of "one type of failures"
. in line 332 - 'eight times on average' in place of "eight time on average"
. in line 350 - 'times' in place of "Times"
. in line 371 - 'three' in place of "there"
. in line 483 - 'in table 3' in place of "in the table 3"
. in line 485 - '80%' in place of "90%"
Please rewrite the sentence "These properties are fulfilled..." in line 302-303, as it does not tell anything.
Comments on the Quality of English LanguageIn general, the quality of English language is quite good. I could identify a few mistakes (and typos) listed in the box "comments and suggestions for authors".
Author Response
Dear Reviewer,
Thank you for giving us the opportunity to submit a revised draft of the manuscript “Modelling the reliability of logistics flows in a complex production system” for publication in the Energies. We appreciate the time and effort that you dedicated to provide feedback on our manuscript. We are grateful for the comments and suggestions that made it possible to introduce valuable improvements to our paper. We have incorporated the suggestions made by the Reviewers. Please see below a point-by-point response to your comments and suggestions.
Reviewer Comments to the Authors:
The authors present a methodology to evaluate the impact of a machine's downtime on the production system. The organization of the manuscript, however, does not explicit the outcomes of the work.
This is because the section "Materials and Methods" includes results that should be placed in another section and this way makes the contributions of the work more clear.
Therefore, I would recommend conveniently splitting section 2 into one or more sections and even merging these new sections with the original sections "Results" and "Discussion".
Also, the section "Results" has some content, e.g., the three areas in which the proposed model can be applied (line 554 to 565), that could integrate the section "Materials and Methods".
In fact, from the sections "Results" and "Discussion" it seems that something is missing in the work, I mean, the objectives were not met. Thus, I would suggest reorganizing the manuscript's structure (as mentioned above) and resetting the aim of the paper.
Authors response: The structure of the article, according to the Reviewer’s recommendation, was revised. Section “Material and Methods” was divided in two sections: “Materials and Methods” and “Methodology for the class of production systems chosen for consideration”. The section dedicated for results was revised: information about areas in which the proposed model can be applied was included in the “Methodology for the class of production systems chosen for consideration” and the precise information about the work objectives was added.
In addition:
Please revise the abbreviations: SME was not defined in line 43; PS in line 91 should be placed between parentheses; TPM, MTTR, MTTF, MTBF were defined twice (in lines 278-281).
Authors response: The abbreviation SME definition was added, PS was placed between parentheses, TPM, MTTR, MTTF, MTBF abbreviations definitions in lines 278-281 were deleted (definitions are included in lines 107-109).
Please correct the following mistakes:
. in line 105 - 'presented' in place of "resented"
. in line 270 - 'number' in place of "n umber"
. in line 297 - 'and' in place of "oraz"
. in line 306 - 'in figure 2' in place of "in the figure 2"
. in line 326 - 'performed' in place of "performer"
. in line 330 - 'one type of failure' in place of "one type of failures"
. in line 332 - 'eight times on average' in place of "eight time on average"
. in line 350 - 'times' in place of "Times"
. in line 371 - 'three' in place of "there"
. in line 483 - 'in table 3' in place of "in the table 3"
. in line 485 - '80%' in place of "90%"
Authors response: The mistakes included in the list were corrected. Thank you for listing them.
Please rewrite the sentence "These properties are fulfilled..." in line 302-303, as it does not tell anything.
Authors response: The sentence: “These properties are fulfilled in the manufacturing the standardized BOM (Bill of Materials) structure components.” was rewritten. The revised version is as follows: “Requirements highlighted in the previous sentence are fulfilled in the analysed company during manufacturing of the standardized BOM (Bill of Materials) components.”.
Reviewer 2 Report
Comments and Suggestions for Authors
The article is relevant, it examines the problems of detecting failures in the delivery of products, analyzes the issues of intra–production logistics - transport flows within the production system. The authors propose a methodology for evaluating production logistics efficiency based on the synthesis of the theory of complex systems and synchronization of production chains.
Comments on the article:
1. There is no definition of the basic concepts and categories used in the article – a complex system, system synchronization, system stability, system survival
2. It is not clear what constituted the empirical basis of the study, where did the data come from?
3. Can the data obtained for this enterprise be applied to other production systems?
4. How appropriate is it to use such complex calculations in an enterprise, is it possible to automate calculations in the form of algorithms?
5. What is the sample size, is it sufficient to talk about the statistical significance of the empirical data obtained?
Author Response
Dear Reviewer,
Thank you for giving us the opportunity to submit a revised draft of the manuscript “Modelling the reliability of logistics flows in a complex production system” for publication in the Energies. We appreciate the time and effort that you dedicated to provide feedback on our manuscript. We are grateful for the comments and suggestions that made it possible to introduce valuable improvements to our paper. We have incorporated the suggestions made by the Reviewers. Please see below a point-by-point response to your comments and suggestions.
Reviewer Comments to the Authors:
The article is relevant, it examines the problems of detecting failures in the delivery of products, analyzes the issues of intra–production logistics - transport flows within the production system. The authors propose a methodology for evaluating production logistics efficiency based on the synthesis of the theory of complex systems and synchronization of production chains.
Comments on the article:
- There is no definition of the basic concepts and categories used in the article – a complex system, system synchronization, system stability, system survival
Authors response: The definitions of basic concepts used in the article were added: definition of a complex production system (lines 121-128), system stability (lines 208-217) and system survival (lines 272-276).
Definition of a single synchronous relation is included in lines 184-192, definition of chain of relations is included in lines 194-200 and synchronous relations chain concept is defined in lines 200-202.
- It is not clear what constituted the empirical basis of the study, where did the data come from?
Authors response: The empirical basis of the study constituted data gathered at production lines of glass processing and sheet metal processing in a manufacturing company.
- Can the data obtained for this enterprise be applied to other production systems?
Authors response: Thank you for this question – in the section 3 (Methodology for the class of production systems chosen for consideration) we supplemented information about possible applications: “The presented methodology has utilitarian properties and can be applied to any class of production systems”. However information about utilitarian properties of the presented methodology and possibility of applying the presented algorithm to any manufacturing system is included in the Introduction section (lines 95-97).
- How appropriate is it to use such complex calculations in an enterprise, is it possible to automate calculations in the form of algorithms?
Authors response: It is possible to automate calculations in the form of algorithms. Work is currently in progress to develop the model mentioned in section 4 (formulas 32, 33). In the next stage, the limits of applicability of the algorithm and the influence of rounding on the size of the error will be verified. Once, these issues have been verified, work on the software algorithm can begin.
- What is the sample size, is it sufficient to talk about the statistical significance of the empirical data obtained?
Authors response: The calculated values of correct operation and failure durations for sheet metal processing line were determined on the basis of empirical data over a six-year period of time (line 380). The glass processing line (as mentioned in the manuscript, lines: 527-528) is a new production unit, the empirical data were collected in a period of one year.
Round 2
Reviewer 1 Report
Comments and Suggestions for Authors
The recommended changes were implemented by the authors.